# THE FAIR LANGUAGE MODEL PARADOX

## ABSTRACT

Large Language Models (LLMs) are widely deployed in real-world applications, yet little is known about their training dynamics at the token level. Evaluation typically relies on aggregated training loss, measured at the batch level, which overlooks subtle per-token biases arising from (i) varying token-level dynamics and (ii) structural biases introduced by hyperparameters. While weight decay is commonly used to stabilize training, we reveal that it silently introduces performance biases detectable only at the token level. In fact, we empirically show across different dataset sizes, model architectures and sizes ranging from 270M to 3B parameters that as weight decay increases, low-frequency tokens are disproportionately depreciated. This is particularly concerning, as these neglected low-frequency tokens represent the vast majority of the token distribution in most languages, calling for novel regularization techniques that ensure fairness across all available tokens.

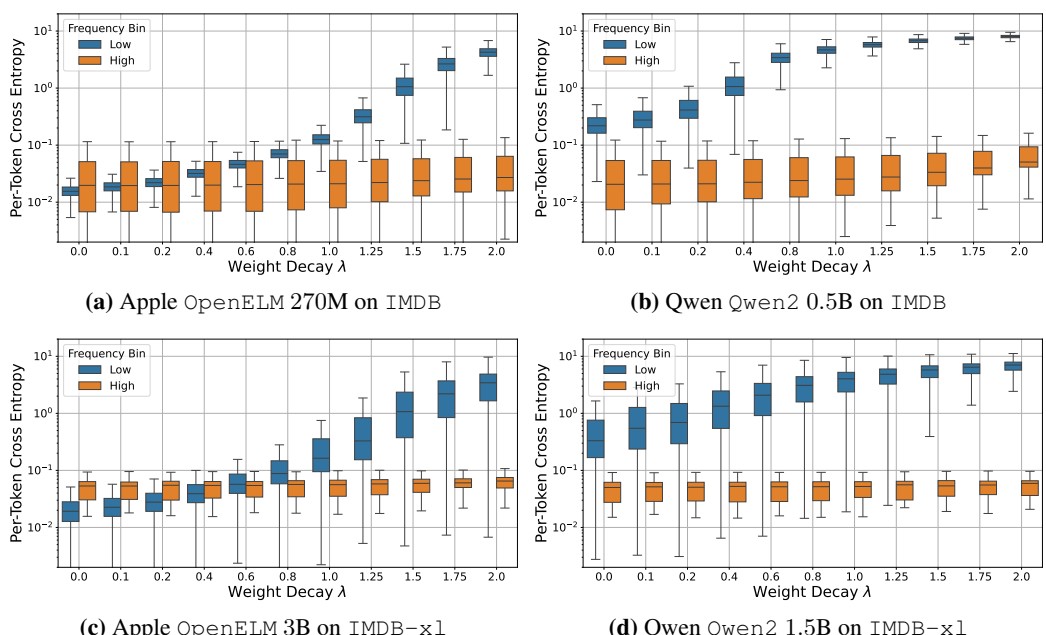

**(a)** Apple `OpenELM` 270M on `IMDB`

**(b)** Qwen `Qwen2` 0.5B on `IMDB`

**(c)** Apple `OpenELM` 3B on `IMDB-xl`

**(d)** Qwen `Qwen2` 1.5B on `IMDB-xl`

Figure 1: We compare the per-token cross-entropy loss for low (blue) and high (orange) frequency tokens when training different LLM architectures and sizes with varying weight decay $\lambda \in (0.0, 2.0)$ on the `IMDB` dataset using a `BPE` tokenizer with a vocabulary size of 32005. As weigth decay increases, the model disproportionately disregards low-frequency tokens, which make up the vast majority of tokens in language datasets. Low-frequency tokens suffer from higher cross-entropy loss, while high-frequency tokens remain largely unaffected. Critically, the degradation of low-frequency token performance happens *silently*, as the average training loss, monitored by practitioners, remains largely unchanged across different levels of weight decay. An example of prompt with segmentation of which tokens are low and high frequency is provided in Figure 2.

> i watched this film for 45 minutes and counted 9 mullets . that 's a mullet every 5 minutes . seriously though , this film is living proof that formula works . if it ain 't broke , it don 't need fix in . a streetwise - yet - v ulner able heroine , a hardened ex - cop martial arts master with a heart of gold and a serial killer with ' iss ues '. pure magic . 

Figure 2: Depiction of a training set prompt from `IMDB` with characters colored by token frequency: low-frequency (blue) and high-frequency (orange). The coloring threshold is 1026 (P99). Tokens appearing fewer than 1026 times in the dataset are blue, otherwise they are colored in orange.

## 1 INTRODUCTION

A major challenge in machine learning is designing algorithms that generalize well from training data. One of the classic methods for promoting generalization (Krogh & Hertz, 1991; Shalev-Shwartz & Ben-David, 2014) is the use of regularization techniques, such as $L_2$ regularization, to limit model complexity. Empirical evidence from classification problems with balanced classes shows that increasing weight decay, while effectively fitting the data and minimizing training loss, generally improves performance on unseen data. However, a recent study by Balestriero et al. (2024) demonstrates that when training classifiers like `ResNet-50` (He et al., 2016) on computer vision tasks such as `ImageNet` classification (Russakovsky et al., 2015), higher weight decay leads to undesired behavior, causing the model to neglect certain classes. This class-dependent effect of regularization is further amplified by imbalanced class distributions, as increasing weight decay does not result in a uniform performance decline across all classes. Instead, the model underperforms on low-probability classes while performing significantly better on more prevalent ones.

This class-dependent behavior is not unique to vision tasks. In Natural Language Processing (NLP), a shift from traditional classification settings has led to a lack of attention toward token frequency and the per-token effects of regularization. In modern large language models (LLMs) trained on text data (Brown et al., 2020; Radford et al., 2018; OpenAI et al., 2024; Anthropic, 2023; Ortiz, 2023), the task of predicting the next token from a large vocabulary also results in significant token frequency imbalances. For instance, as shown in Figures 3, in the `IMDB` dataset (Maas et al., 2011), 95% of the total tokens in the data are captured by the top 0.01% of tokens. This indicates that the vast majority of tokens appear infrequently, while a small set of tokens dominates, creating a substantial imbalance. Additionally, the proportion of low-frequency tokens tends to increase as the vocabulary expands. This raises a critical question:

> *Can regularization techniques, such as weight decay, typically used to promote generalization, ensure fairness across tokens when applied to LLMs trained on imbalanced token distributions?*

**Contributions.** In order to study that question, we investigate the influence of weight decay on the token-level prediction performance of large language models, uncovering critical insights that are often overlooked when relying solely on aggregated performance metrics. Our study provides several key contributions:

- The next-token classification task suffers from severe class imbalance. Figures 3 demonstrate that the class distribution follows a heavy-tailed pattern, with the vast majority of classes being low-frequency and only a small portion being high-frequency.
- We trained the Apple `OpenELM` models with 270M and 3B parameters, as well as `Qwen2` models with 0.5B and 1.5B parameters, on both the `IMDB` dataset and its extended version `IMDB-xl`, using varying levels of weight decay that yielded acceptable training losses. As observed in Figure 1, the models' performance on low-frequency tokens significantly degrades as weight decay increases.
- We observe that higher-frequency tokens are consistently learned faster than low-frequency tokens across multiple random seeds, with the gap in learning speed widening as weight decay increases, suggesting that regularization may disproportionately disadvantage rare tokens.

These findings expose a *critical dilemma*. Practitioners often use aggressive weight decay to train LLMs—intended to stabilize training—but unintentionally and silently degrade the model's performance on low-frequency tokens, which make up the majority of the data. While conventional wisdom

advocates for increased weight decay to promote generalization, in language modeling, this strategy results in the unintended consequence of "neglecting" low-frequency tokens. This introduces harmful biases, favoring more common tokens, often reflecting the language patterns of majority groups. Our results emphasize the need for regularization techniques that explicitly address token imbalances.

## 2 RELATED WORK

**Weight Decay and Generalization.** $L_2$ regularization was initially introduced to stabilize solutions to ill-posed problems (Tikhonov, 1943) and later adopted to enhance the generalization of neural networks (Krogh & Hertz, 1991). While many studies have linked low-norm solutions to improved generalization (Neyshabur et al., 2015; Golowich et al., 2020; Bartlett & Mendelson, 2001; Bartlett et al., 2017; Arora et al., 2018; Galanti et al., 2023c; Wei & Ma, 2019; Li et al., 2018), the precise relationship between $L_2$ regularization and generalization remains a topic of debate. Several works (Zhang et al., 2017; Jiang* et al., 2020) argue that norm-based measures alone are insufficient to fully explain generalization in deep learning. For example, Zhang et al. (2017) found that although weight decay can improve test accuracy, the overall effect is typically modest—around $1 - 2\%$ on `ImageNet`. Nonetheless, other studies have demonstrated that weight decay helps alleviate (Nakkiran et al., 2021; Pezeshki et al., 2022) the double descent phenomenon (Belkin et al., 2019; Nakkiran et al., 2020) and is critical for achieving Grokking in mathematical reasoning tasks (Power et al., 2022; Varma et al., 2023).

**Weight Decay, Optimization, and Inductive Biases.** Despite its modest impact on generalization, weight decay is widely employed in many state-of-the-art language models, including `GPT-3` (Brown et al., 2020), `Chinchilla` (Hoffmann et al., 2024), and `LLaMA` (Touvron et al., 2023a;b; Dubey et al., 2024). These models are typically trained using a "one-pass" stochastic gradient descent (`SGD`) regime, where the optimizer directly minimizes the population error.

As shown in (Andriushchenko et al., 2023), the training and validation losses remain closely aligned across different levels of weight decay. While weight decay's effect on generalization is limited, it plays a crucial role in improving optimization. For example, both the `Chinchilla` paper (Hoffmann et al., 2024) (see Figure A7) and (Andriushchenko et al., 2023) (see Figure 4) demonstrate that weight decay in `AdamW` leads to lower training loss compared to `Adam`, particularly toward the end of training. Other studies (van Laarhoven, 2017; Zhang et al., 2019; Li & Arora, 2019; Li et al., 2020; Lewkowycz & Gur-Ari, 2020) have shown that weight decay enhances training stability by controlling the "effective learning rate" in scale-invariant neural networks. Additionally, other works (Galanti et al., 2023b; Rangamani & Banburski-Fahey, 2022; Pan & Cao, 2024; Beneventano et al., 2024) reveal that weight decay contributes to various inductive biases, such as rank minimization and neural collapse, which are beneficial for network compression (Denton et al., 2014; Alvarez & Salzmann, 2017; Tukan et al., 2021; Yu et al., 2017) and downstream performance (Galanti et al., 2022; 2023a).

**Training with Imbalanced Classes and Minority Collapse.** Training with imbalanced classes presents a significant challenge in machine learning. Empirical studies consistently show that the weight vectors associated with the more frequent classes tend to have larger norms, which pushes the decision boundary toward the minority classes. As a result, the feature space allocated to less frequent classes shrinks, leading to a notable drop in performance (Kim & Kim, 2020; Kang et al., 2019; Cao et al., 2019; Ye et al., 2020; Liu et al., 2023; Kang et al., 2020; Balestriero et al., 2022). For instance, (Balestriero et al., 2022) showed that when training neural networks for classification of visual data, higher levels of weight decay introduce a stronger bias for the model to prioritize higher-probability classes over lower-probability ones.

To gain deeper insights into this issue, several works have investigated this phenomenon from a theoretical perspective. Fang et al. (2021) proposed the unconstrained features model (UFM) as a simplified framework for exploring the geometric properties of the global minima in cross-entropy loss with regularization, particularly in overparameterized neural networks. In the case of a balanced dataset, they demonstrated that neural collapse (NC) occurs at any global minimizer of the loss function combined with regularization. However, in the case of class imbalance, neural networks exhibit distinct geometric patterns, and some of the NC properties no longer hold (Dang et al., 2023; Thrampoulidis et al., 2022; Hong & Ling, 2023; Dang et al., 2024). While last-layer features for samples in the same class still collapse to their respective class means (NC1), the class means and classifier weights no longer form a Simplex Equiangular Tight Frame (ETF), violating NC2 (Fang

et al., 2021). In more extreme cases, when the imbalance becomes severe, the classifier weights for minority classes can collapse onto each other, rendering them indistinguishable from other classes (Fang et al., 2021). This phenomenon, referred to as "Minority Collapse," explains the sharp decline in accuracy for minority classes in imbalanced settings.

Building on these finding, we explore whether similar behavior arises in next-token prediction tasks, where the problem can be viewed as a (very noisy) classification task, with the next token acting as the "class" for a given sequence.

## 3 EXPERIMENTAL SETTINGS

**Problem Setup.** We focus on the task of next-token prediction in autoregressive language modeling, a self-supervised learning problem central to natural language processing. Given a sequence of tokens $x_i = (x_{i,1}, ..., x_{i,n})$, the objective is to model the conditional probability distribution $p(x_{i,t}|x_{i,<t})$ for each token position $t$. Formally, let $\mathcal{D} = \{x_i\}_{i=1}^m$ be a text corpus, where each $x_i = (x_{i,1}, ..., x_{i,n}) \in \mathcal{S} = \mathcal{V}^n$ is a sequence of $n$ tokens, and $x_{i,t} \in \mathcal{V}$ belongs to a fixed vocabulary $\mathcal{V}$ of size $V$. We train different transformer-based models $f_\theta : \mathcal{S} \to [V]$ mapping from sequence to logits to minimize the regularized empirical risk:

$$L_{\mathcal{D}}^\lambda(f) = \frac{1}{m} \sum_{i=1}^m \left( \frac{1}{n-1} \sum_{t=1}^{n-1} \ell(f_\theta(x_{i,\leq t}), x_{i,t+1}) \right) + \lambda ||\theta||_2^2,$$

where $\ell$ denotes the cross-entropy loss, $\lambda$ is the weight decay coefficient, $||\theta||_2$ is the $L_2$ norm of the model parameters, $x_{i,\leq t}$ the sequence up to position $t$ in the $i$-th sample, and $x_{i,t+1}$ is the next token to be predicted.

**Model Architecture.** For our experiments, we train multiple models, including the Apple `OpenELM` models with 270M and 3B parameters (Mehta et al., 2024), as well as the Qwen `Qwen2` models with 0.5B and 1.5B parameters (Yang et al., 2024). The small models ($< 1$B parameters) were configured with a context length of 128 whereas the large ones with a context length of 64. These architectures's moderate size allows us to conduct multiple training runs with different $\lambda$ values, enabling a comprehensive exploration of weight decay's impact on token-level dynamics across various regularization choices.

**Dataset and Tokenizer.** We trained our models on the `IMDB` dataset (Maas et al., 2011), a widely used benchmark due to its balanced sentiment-labeled data. The `IMDB` training split contains 25000 samples. For the scope of this study, we discarded the labels, focusing solely on the raw text data to analyze the impact of hyperparameters on token-level generation performance. Additionally, we created an extended version of the dataset, termed `IMDB-xl`, by incorporating all the unsupervised samples from `IMDB`, which increased the training set to a total of 75000 samples. We used Byte Pair Encoding (`BPE`) (Gage, 1994; Sennrich et al., 2016) as our tokenization method, training a tokenizer on the `IMDB` dataset's training set with a target vocabulary size of 32005 tokens. `BPE` ensures that both frequent and infrequent tokens are well represented, providing a suitable basis for analyzing token-level learning. This choice allows us to examine how different tokens, particularly rare ones, are influenced by various weight decay values throughout the training process.

**Hyperparameters and Training.** We conducted experiments across a range of weight decay values $\lambda \in (0.0, 2.0)$. To account for the stochastic nature of training, each configuration was run with the same 5 different random seeds, with results averaged and presented with confidence intervals. The models were trained using the `AdamW` optimizer (Loshchilov & Hutter, 2019) to decouple the weight decay from gradient updates, with a learning rate of $5e-5$ and a cosine decay schedule. A warm-up period of 10 training steps was used for stabilization during early training. We trained each LLM for a total of 10000 steps, evaluating token-level metrics every 100 steps to understand how different tokens are learned and represented under varying weight decay values. For small models (less than 1B parameters), we used a batch size of 64 per device on a single NVIDIA `A100` 32GB GPU, whereas for large models, we used a batch size of 16 with a gradient accumulation step of 4 on a single NVIDIA `A100` 80GB GPU. Additionally, mixed precision training (`fp16`) was employed to optimize GPU memory usage, reduce computational load, and accelerate convergence.

**Evaluation Metrics.** Our evaluation used token-level metrics to assess how individual tokens were generated under varying weight decay configurations. Unlike traditional metrics like perplexity, we

focused on whether specific tokens were under-represented during inference, revealing how certain configurations induce bias in token representation, even when aggregate performance appears stable. The *average training loss*, computed with cross-entropy loss over all tokens in a batch, is given by:

$$\mathcal{L}_{\text{avg}} = -\frac{1}{BC} \sum_{b=1}^{B} \sum_{c=1}^{C} \sum_{v=1}^{V} y_{b,c,v} \log p_{b,c,v},$$

where $y_{b,c,v}$ and $p_{b,c,v}$ are the ground truth and predicted probabilities, respectively. This method implicitly favors frequent tokens, as they appear more often, leading the model to prioritize them over low-frequency tokens. To counter this imbalance, we also used a *token-balanced training loss*, ensuring each token contributes equally, regardless of frequency. First, we compute the cross-entropy loss for each token $\ell_{b,c} = -\sum_{v=1}^{V} y_{b,c,v} \log p_{b,c,v}$. We then average these losses by token type, yielding the final token-balanced loss:

$$\mathcal{L}_{\text{tok-bal}} = \frac{1}{V} \sum_{v=1}^{V} \frac{1}{|\{(b,c) : y_{b,c,v} = 1\}|} \sum_{\{(b,c):y_{b,c,v}=1\}} \ell_{b,c}.$$

This approach ensures that low-frequency tokens contribute equally to the loss, leading to a more balanced optimization process, though it may challenge the model's handling of rare tokens, especially under strong regularization. The computation of per-token metrics is summarized in Algorithm 1.

**Per-Token Learning Speed.** To quantify how quickly the model learns individual tokens during training, we introduce the per-token learning speed metric, which measures how rapidly the model minimizes the cross-entropy loss for each token. Specifically, we compute the area under the curve (AUC) of the token's normalized loss trajectory over time, where a smaller AUC indicates faster learning. For each token, we normalize its cross-entropy losses across all training steps to the range $[0, 1]$, ensuring comparability between tokens with different loss scales. Let $\ell_t$ represent the cross-entropy loss for a token at training step $t$, and let $\mathcal{L} = \{\ell_1, \ell_2, \ldots, \ell_T\}$ be the set of losses over $T$ steps. We define the normalized loss at each step as: $\tilde{\ell}_t = \frac{\ell_t - \min(\mathcal{L})}{\max(\mathcal{L}) - \min(\mathcal{L})}$. The area under the normalized loss curve is calculated as: $\text{AUC}(\mathcal{L}) = \int_0^T \tilde{\ell}_t dt$. The learning speed $S$ is then defined as: $S = 1 - \frac{\text{AUC}(\mathcal{L})}{T}$. This ensures that $S$ ranges from 0 to 1, with higher values indicating faster learning. If the range of losses is zero (i.e., $\max(\mathcal{L}) = \min(\mathcal{L})$), we set $S$ to zero, as no learning occurs.

## 4 EMPIRICAL RESULTS

### 4.1 TEXTUAL DATA IS HIGHLY IMBALANCED

As a preliminary step in our study, we investigated how imbalanced textual data actually is. For this purpose we conducted several statistical evaluations of the IMDB dataset in order to verify that indeed the vast majority of tokens are of low-frequency and a small minority of the tokens appear very often in the data as predicted by the Zipf law.

**Token Frequency Percentiles.** Figure 3(a) highlights the extreme token frequency imbalance in the IMDB dataset, tokenized using a BPE tokenizer with a vocabulary size of 32005. The vast majority of the token frequency mass is concentrated in a very small portion of high-frequency tokens. Specifically, 95% of the total tokens in the data is captured by the top 0.01% of tokens, which demonstrates the steep distribution of token frequencies, where very few tokens dominate. To illustrate the calculation: suppose we have a dataset with 100 samples, each consisting of 10 tokens. If a set of 1% of the tokens in the vocabulary appear 800 times across those 1000 tokens, we say that 80% of the total tokens in the data is captured by the top 1% of tokens. Figure 3(b) further emphasizes this imbalance, showing how the proportion of low-frequency tokens (those below the 95th percentile in the data) grows as the vocabulary size increases. This suggests that token imbalance is inherent to language and further amplified as the vocabulary size expands with larger tokenizers.

### 4.2 PERFORMANCE IMPAIRMENT IN LOW-FREQUENCY TOKENS

**Average vs. Token-Balanced Training Loss.** To highlight the effect of weight decay on low-frequency tokens, we compared the average training loss with the token-balanced training loss.

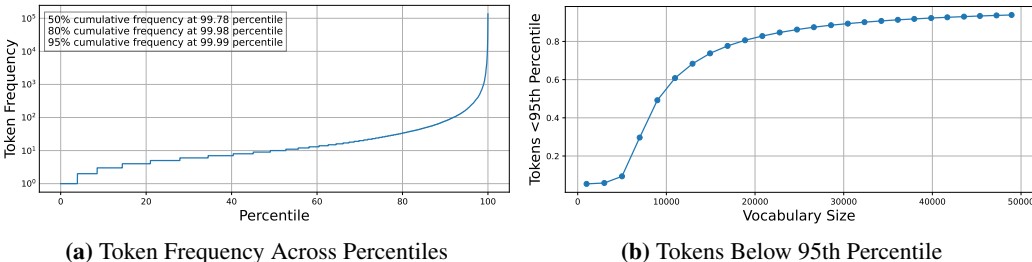

**(a)** Token Frequency Across Percentiles      **(b)** Tokens Below 95th Percentile

Figure 3: Comparison of token frequency distribution and the ratio of low-frequency tokens across varying vocabulary sizes for the IMDB dataset. The left plot shows the token frequency distribution with cumulative frequency thresholds (50%, 80%, and 95%) marked. The right plot illustrates how the ratio of tokens below the 95th percentile increases with vocabulary size, converging to $\approx 0.85$.

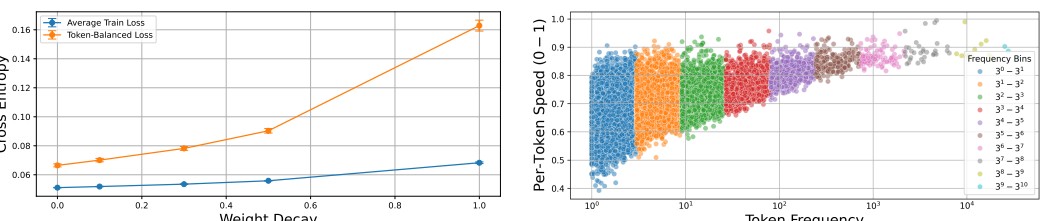

Figure 4: **Impact of Weight Decay on Cross-Entropy.** Average training loss (blue) and class-balanced loss (orange) increase with weight decay. The class-balanced loss is more sensitive due to its focus on low-frequency tokens.

Figure 5: **Token Learning Speed.** Token learning speed $(0-1)$ plotted against frequency (log-scale) for $\lambda = 1.0$. Colors represent token groups by frequency bins, highlighting variation across token frequencies.

Figure 4 presents both metrics at the end of training for models trained with varying weight decay values ($\lambda \in \{0.0, 0.1, 0.3, 0.5, 1.0\}$). As shown, the token-balanced loss rises sharply with increasing weight decay, while the average training loss shows only a slight increase as $\lambda$ moves from $0.0$ to $1.0$. This discrepancy occurs because weight decay disproportionately affects low-frequency tokens, which are given equal importance in the token-balanced loss. In contrast, the average training loss places much less emphasis on low-frequency tokens, making the effects of weight decay less noticeable. For exact values, including per-token perplexity and training accuracy, see Table 1.

**Per-Token Performance vs. Weight Decay.** Beyond comparing the average and token-balanced training losses, we also investigated how increasing weight decay influences performance on low- and high-frequency tokens separately. In this experiment, we examine how increasing weight decay affects the per-task loss function for tokens of different frequencies. To this end, we trained multiple versions of the same model (e.g., Apple `OpenELM` 270M) with varying degrees of weight decay on a given dataset (e.g., `IMDB`) and compared the average loss function for low-frequency and high-frequency tokens. As shown in Figure 1, the performance on high-frequency tokens remains largely unaffected by the increase in weight decay, in contrast to the loss for low-frequency tokens, which increases significantly with higher weight decay. Here, low-frequency tokens are those that appear between $3^0$ and $3^1$ times as the next token in the data, while the highest-frequency tokens appear between $3^9$ and $3^{10}$ times as the next token.

**Per-Token Performance vs. Frequency.** As a next step, we would like to visualize the distribution of per-token losses and accuracies across tokens of varying frequencies. In Figure 6, tokens are grouped into bins, where the $i$th bin contains tokens with frequencies between $3^{i-1}$ and $3^i$. For each token, we plotted in Figure 6 (a) its average per-token cross-entropy and in Figure 6 (b) the per-token accuracy, averaging over all sequences where that token is predicted and over all random seeds. As shown, model performance improves significantly and consistently as token frequency increases, with high-frequency tokens benefiting the most. In Figure 5, we replicated this analysis for per-token learning speed, which also monotonically improves as token frequency increases, following the same pattern observed for loss and accuracy.

| Weight Decay $\lambda$ | 0.0 | 0.1 | 0.3 | 0.5 | 1.0 |
|---|---|---|---|---|---|
| Training Loss | 0.051±0.000 | 0.052±0.000 | 0.054±0.000 | 0.056±0.000 | 0.068±0.001 |
| Per-Token Loss | 0.066±0.001 | 0.070±0.001 | 0.078±0.001 | 0.090±0.002 | 0.163±0.004 |
| Per-Token PPL | 1.069±0.003 | 1.073±0.003 | 1.081±0.003 | 1.095±0.003 | 1.177±0.003 |
| Per-Token Accuracy (%) | 98.798±0.031 | 98.781±0.029 | 98.778±0.032 | 98.759±0.028 | 98.714±0.029 |

Table 1: Impact of weight decay ($\lambda$) on model performance metrics (mean±std).

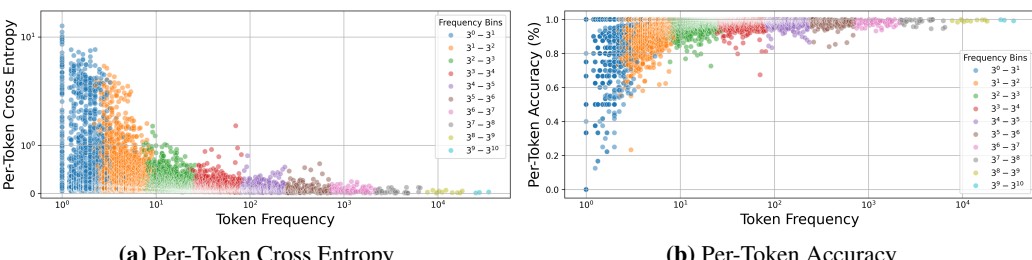

**(a)** Per-Token Cross Entropy        **(b)** Per-Token Accuracy

Figure 6: **(a)** The relationship between token frequency (log-scale) and per-token cross entropy (symlog-scale) across frequency bins, representing powers of 3. Lower-frequency tokens exhibit significantly higher cross entropy, indicating weaker learning. **(b)** The relationship between token frequency and accuracy across frequency bins. Higher-frequency tokens achieve higher accuracy, while lower-frequency tokens demonstrate more variability in performance. Both plots are $\lambda = 1.0$.

**High-Frequency Tokens are Learned Faster.** In Figure 7 (b), tokens are divided into bins, selected in the same way as before, to compare the average token learning speed (as defined earlier) and its standard deviation for two models: one trained without weight decay ($\lambda = 0$, blue) and the other with weight decay ($\lambda = 1$, orange). The percentage of tokens in each bin is also shown, with the majority concentrated in the low-frequency bins. As observed, less frequent tokens are learned more slowly by both models. Notably, the gap in learning speed between the models widens for lower-frequency tokens but diminishes for higher-frequency tokens. This indicates that increasing weight decay disproportionately deprioritizes low-frequency tokens.

## 5 THEORETICAL DISCUSSION

It is well known that the class frequency is positively correlated with the norm of the top layer classifier of the given class (Kang et al., 2019; Huang et al., 2016; Kim & Kim, 2020). For instance, Dang et al. (2024) considered a variant of unconstrained features model (UFM) (Fang et al., 2021), in which the features are constrained to be non-negative, motivated by the fact that features are usually the output of ReLU activations in many common architectures. We use their framework to analyze the influence of the token frequency and the weight decay on the per-token loss function. Formally, suppose we have a set of possible tokens $\mathcal{V}$ (one-hot encodings of the numbers in $[V]$) and a dataset $\mathcal{D} = \cup_{k=1}^{V}\{x_{k,i}\}_{i=1}^{n_k}$ of sequences $x_{k,i} = (x_{k,i,1}, \ldots, x_{k,i,n})$ of tokens with the next token being $k \in [V]$ (whose one-hot encoding is $y_k$). Within the model proposed in (Dang et al., 2024), for each sequence $x_{k,i}$ we learn an unconstrained feature representation $h_{k,i} \in \mathbb{R}^d$ together a linear layer $W \in \mathbb{R}^{V \times d}$ using the Cross-Entropy loss:

$$\min_{W,H} \quad \frac{1}{N}\sum_{k=1}^{V}\sum_{i=1}^{n_k}\ell_{\text{CE}}(Wh_{k,i}, y_k) + \frac{\lambda_W}{2}\|W\|_F^2 + \lambda_H\|H\|_F^2, \tag{1}$$

$$\text{s.t.} \quad H \geq 0, \ \lambda_W > 0, \ \lambda_H > 0,$$

where $\ell_{\text{CE}}(z, y_k) = -\log\left(\frac{\exp(z_k)}{\sum_{i=1}^{V}\exp(z_i)}\right)$, $H := [h_{1,1}, \ldots, h_{1,n_1}, h_{2,1}, \ldots, h_{V,n_V}] \in \mathbb{R}^{d \times N}$ (where $N = \sum_{k=1}^{V} n_k$) are the learned representations for each sample $(k, i)$ and $H \geq 0$ denotes entry-wise non-negativity. In addition, $W = [w_1, w_2, \ldots, w_V]^\top \in \mathbb{R}^{V \times d}$ to be the last-layer weight matrix, with $w_k \in \mathbb{R}^d$ being the $k$-th row of $W$.

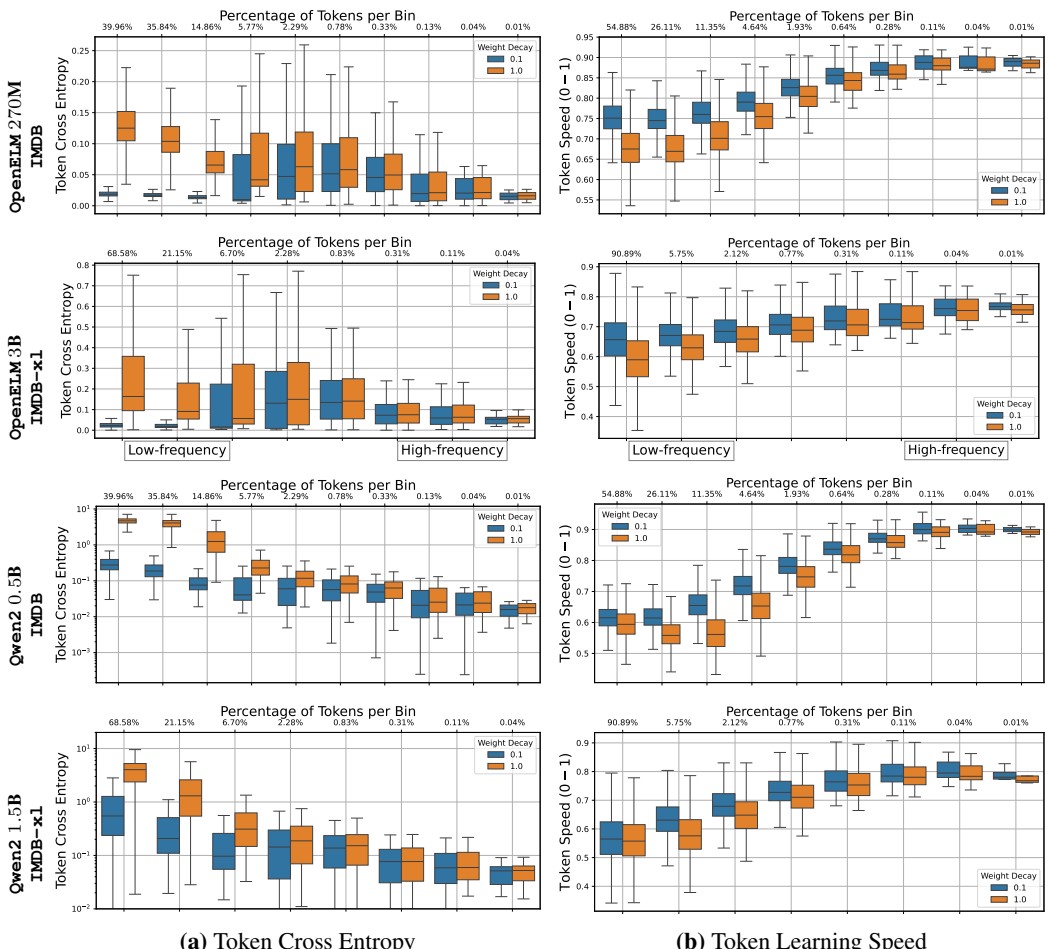

**(a)** Token Cross Entropy       **(b)** Token Learning Speed

Figure 7: We compare the **(a)** per-token cross-entropy loss and **(b)** learning speed across token frequencies when training an LLM with weight decay $\lambda = 0.1$ and $\lambda = 1.0$, on the `IMDB` and `IMDB-xl` datasets using a `BPE` tokenizer with a vocabulary size of 32005. As weight decay increases, the model disproportionately disregards low-frequency tokens, which make up the vast majority of tokens in language datasets. Low-frequency tokens suffer from higher cross-entropy loss and reduced learning speed, while high-frequency tokens remain largely unaffected. Critically, the degradation of low-frequency token performance happens *silently*, as the average training loss, monitored by practitioners, remains largely unchanged across different levels of weight decay.

While the above formulation does not exactly match the practice (since $H$ is a matrix of free parameters instead of the outputs of a neural network), this abstraction can help understand how the per-token frequency $n_k$ and the regularization parameters $\lambda_W$ and $\lambda_H$ influence the per-token loss and classifier. For instance, according to Theorem 4.1 and Proposition 4.3 in (Dang et al., 2024), if $d \geq V$ then any global minimum of Eq. 1 satisfies:

    (a) Within-class feature collapse: $\forall\, k \in [V],\ i \neq j \in [n_k]:\ h_{k,i} = h_{k,j} = \mu_k$.

    (b) Class-mean orthogonality: $\forall\, k \neq l:\ \mu_k^\top \mu_l = 0$.

    (c) Class-mean norm: $\|\mu_k\|^2 = \sqrt{\frac{\lambda_W (V-1)}{\lambda_H V n_k}} M_k$.

    (d) Weight norms: $\|w_k\|^2 = \sqrt{\frac{\lambda_H}{\lambda_W V^3 (V-1)}} \left( (V-1)^2 \sqrt{n_k} M_k + \sum_{j=1}^{V} \sqrt{n_j} M_j \right)$.

Here, $M_k = \left[\log\left((V-1)\left(\frac{\sqrt{n_k}}{N\sqrt{\frac{V-1}{V}}\lambda_W\lambda_H} - 1\right)\right)\right]_\diamond$, where the function $[x]_\diamond$ returns $x$ if $x$ is defined and is positive and $0$ otherwise.

The above series of observations imply that $\mu_k = 0$ if and only if $M_k = 0$. As a result (summarized in Corollary 4.6 in (Dang et al., 2024)), if $n_k \leq \lambda_W \lambda_H N^2 \frac{V-1}{V}$ (which implies $M_k = 0$) the model avoids learning the $k$th token. In particular, ***the number of tokens that are neglected by the model increased when increasing the level of weight decay***.

**Proposition 5.1.** *Suppose $d \geq V$, then any global minimizer $(W, H)$ of the problem obeys* $\ell_{\text{CE}}(Wh_{k,i}, y_k) = \log\left(\sum_{j=1}^{V} \exp\left(\frac{M_j}{V^2}\right)\right) - M_k.$

We observe that the loss function $\ell_{ij} = \ell_{\text{CE}}(Wh_{k,i}, y_k)$ can be decomposed into two components: the first part, $\ell'$, is independent of both $k$ and $i$, while the second part depends on $-M_k$. Consequently, tokens $k$ associated with larger values of $M_k$ incur a smaller per-token loss. Since $M_k$ increases with $n_k$, it follows that ***the per-token loss is smaller for tokens of higher frequency***. This aligns with the results in Fig. 7 (a), where the loss function decreases monotonically for higher-frequency tokens when training with weight decay.

Now, suppose we set $\lambda_W = \lambda_H = \lambda$. The derivative of the per-token loss $\ell_{k,i}$ with respect to $\lambda$ is given by: $\frac{\partial \ell_{k,i}}{\partial \lambda} = \frac{\partial \ell'}{\partial \lambda} + \left(\frac{1}{\lambda} + \frac{N\sqrt{V-1}}{\sqrt{n_k V} - \lambda N\sqrt{V-1}}\right) \cdot \mathbb{I}\left[n_k > \lambda^2 N^2 \frac{V-1}{V}\right]$. We note that the first term of the derivative, $\frac{\partial \ell'}{\partial \lambda}$, is independent of $n_k$, while the second term is monotonically decreasing with respect to $n_k$, provided that $n_k > \lambda^2 N^2 \frac{V-1}{V}$ ($k$ is a non-neglected token). Therefore, the derivative of the loss for non-neglected tokens $k$ is higher for smaller values of $n_k$. In particular, ***the loss for low-frequency tokens grows at a faster rate compared to high-frequency tokens when increasing $\lambda$***. This can be observed in Fig. 7 (a) and Fig. 4, where, although the losses generally increase across all types of tokens as a function of $\lambda$, they increase more significantly for low-frequency tokens.

As a final note, although one might think that the observations made in Sec. 4 could be due to poor training, these theoretical results emphasize that they are actually caused by a fundamental issue when training next-token predictors. The results above apply to the global minima of the objective, indicating that they pertain to situations where the training was in fact optimal.

## 6 CONCLUSION

We investigated the impact of weight decay on token-level learning dynamics in large language models. Our findings reveal critical insights into how weight decay affects the learning process of individual tokens—effects that are hidden when relying solely on aggregated metrics. We demonstrated that increasing weight decay disproportionately harms the performance of low-frequency tokens, even when the overall average loss remains largely unchanged. Additionally, we observed that higher-frequency tokens are generally learned faster than their low-frequency counterparts. This interplay between token frequency, performance, and regularization highlights the nuanced effects of training techniques on different parts of the model's vocabulary.

These results expose a significant oversight in current LLM training practices. Weight decay is commonly employed to reduce overfitting and enhance optimization. While this seems beneficial at first—due to improved convergence and stability in overall loss metrics—our analysis uncovers a hidden pitfall: weight decay can severely compromise the model's ability to handle low-frequency tokens. Crucially, this degradation goes unnoticed when only aggregated loss metrics are considered. This discrepancy between aggregated performance and token-specific learning underscores the need for fine-grained, token-level evaluations. Without such assessments, models risk sacrificing performance on rare or specialized vocabulary, potentially limiting their effectiveness in domains that require precise handling of low-frequency terms.

As illustrated in Figure 3, the imbalance becomes more pronounced as vocabulary sizes increase. This is particularly concerning as vocabulary sizes are continuously expanded to improve model performance and broaden capabilities (Takase et al., 2024; Tao et al., 2024; Toraman et al., 2023). For example, while LLaMA-v1 (Touvron et al., 2023a) and LLaMA-v2 Touvron et al. (2023b) used a vocabulary of 32000 tokens, LLaMA-v3 (Dubey et al., 2024) expanded this to 128256 tokens,

`Qwen2` (Yang et al., 2024) further extended it to 151936 tokens, and `Gemma-2` (Team et al., 2024) increased the size to 256128 tokens.

**Broader Impact.** Our work enhances understanding of how weight decay affects token-level performance in LLMs while remaining undetected by aggregated metrics. This is crucial for ensuring fairness and reliability in real-world applications. While our societal impact is indirect, it underscores the need for more granular evaluation metrics to detect and mitigate potential biases on low-frequency tokens. These findings contribute to the development of more equitable and robust AI systems.

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

## A  PROOF OF PROPOSITION 5.1

**Proposition 5.1.** *Suppose $d \geq V$, then any global minimizer $(W, H)$ of the problem obeys*
$$\ell_{\text{CE}}(Wh_{k,i}, y_k) = \log\left(\sum_{j=1}^{V} \exp\left(\frac{M_j}{V^2}\right)\right) - M_k.$$

*Proof.* By combining (a-d) in Theorem 4.1 in (Dang et al., 2024), we have:

$$w_k^\top h_{k,i} = w_k^\top \mu_k$$

$$= \sqrt{\frac{\lambda_H}{\lambda_W V(V-1)}} \left( V\sqrt{n_k}\mu_k^\top \mu_k - \sum_{m=1}^{V} \sqrt{n_m}\mu_m^\top \mu_k \right)$$

$$= \sqrt{\frac{\lambda_H}{\lambda_W V(V-1)}} \left( V\sqrt{n_k}\sqrt{\frac{V-1}{V}\frac{\lambda_W}{\lambda_H}\frac{1}{n_k}}M_k \right)$$

$$= M_k$$

Similarly, we can show that $w_j^\top h_{k,i} = \frac{M_j}{V^2}$. Hence, the loss function is equal to:

$$\ell_{\text{CE}}(Wh_{k,i}, y_k) = -\log\left(\frac{\exp(w_k^\top h_{k,i})}{\sum_{j=1}^{V}\exp(w_j^\top h_{k,i})}\right) = \log\left(\sum_{j=1}^{V}\exp\left(\frac{M_j}{V^2}\right)\right) - M_k.$$

$\square$

## B  PER-TOKEN METRICS

This appendix presents Algorithm 1, which details our method for computing per-token metrics used throughout this paper. This algorithm is central to our analysis, enabling the calculation of fine-grained token-level performance measures that underpin our study's findings.

---

**Algorithm 1:** Token-level Metric Computation Algorithm

---

**Input:** Logits $\mathbf{L}$, Labels $\mathbf{y}$, Tokenizer $\mathcal{T}$
**Output:** Token-level metrics $\mathcal{M}$

1 Initialize token metrics dictionary $\mathcal{M}$;
2 **foreach** *step* **do**
3    Get logits $\mathbf{L}$ and labels $\mathbf{y}$;
4    Shift logits $\mathbf{L}' \leftarrow \mathbf{L}[:, :-1]$;
5    Shift labels $\mathbf{y}' \leftarrow \mathbf{y}[:, 1:]$;
6    Compute predictions $\hat{\mathbf{y}} \leftarrow \arg\max(\mathbf{L}')$;
7    **foreach** *token $t$ in $\mathbf{y}'$* **do**
8       Compute per-token loss $\ell_t \leftarrow \text{CrossEntropyLoss}(\mathbf{L}_t, \mathbf{y}'_t)$;
9       Update metrics $\mathcal{M}[t] \leftarrow \mathcal{M}[t] + \{\text{loss} : \ell_t, \text{correct} : (\hat{\mathbf{y}}_t = \mathbf{y}'_t)\}$;
10    **end**
11 **end**

---

## C  IMDB TOKEN DISTRIBUTION

As an additional evaluation, we examine the token frequency distribution of the `IMDB` dataset, as shown in Figure 8. Using a `BPE` tokenizer with a 32005-token vocabulary, we illustrate the distribution of high-frequency and low-frequency tokens. The histogram reveals a striking imbalance typical of natural language datasets: a few tokens occur extremely frequently, while most unique tokens appear rarely. This long-tail distribution is a fundamental characteristic of language data.

## D  CLASS IMBALANCE IN VISION

In order to show that the behavior reported in the main text is not specific to LLMs but holds also for other types of domains, we experimented with image classification. We present an analysis of

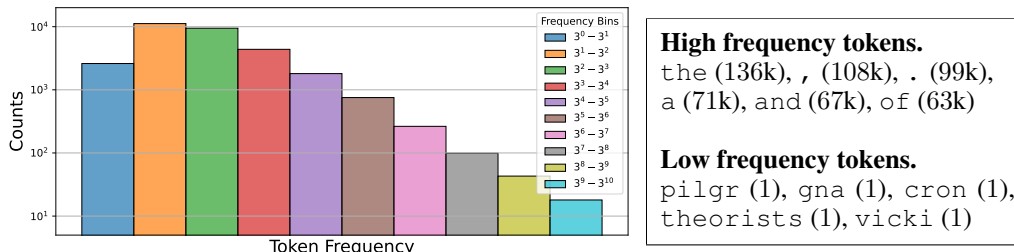

Figure 8: **Token frequency distribution of the `IMDB` dataset** Maas et al. (2011) using a `BPE` tokenizer (vocab size 32005). y-axis: number of unique tokens per frequency bin. x-axis: token frequency bins (logarithmic scale). Rightmost bin shows 3 tokens (`the`, punctuation) appearing $3^9$ to $3^{10}$ times each. This highlights high-frequency tokens (`the`, punctuation) and rare tokens ("`pilgr`", "`cron`") in lower bins, illustrating the long-tail distribution typical in natural language datasets.

how dataset balance and weight decay affect the performance of a `ResNet9` model (He et al., 2016) trained for `CIFAR10` (Krizhevsky & Hinton, 2009) classification with imbalanced classes. This highlights the interplay between data distribution, regularization, and model performance. To obtain the imbalanced CIFAR10 dataset, we target a number of samples per class of [91, 142, 222, 347, 541, 845, 1317, 2055, 3205, 5000]. We randomly subsampled the corresponding classes in order to obtain those number of images per class.

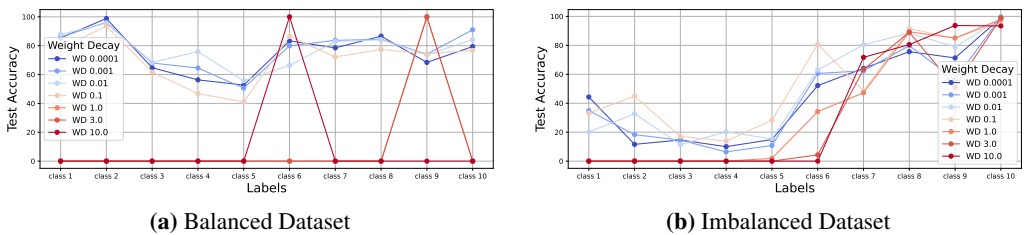

**(a)** Balanced Dataset          **(b)** Imbalanced Dataset

Figure 9: Performance comparison of `ResNet9` on balanced (left) and imbalanced (right) `CIFAR10` datasets. The x-axis represents the 10 `CIFAR10` classes, while the y-axis shows the test accuracy for each class. Different lines correspond to various weight decay values used during training. In the balanced dataset, performance is relatively uniform across classes, with overall homogenous degradation at high weight decay. The imbalanced dataset reveals a clear trend where less frequent classes perform worse, with this effect exacerbated by stronger weight decay values.

