# OpenReview forum: "The Fair Language Model Paradox"
_ICLR.cc/2025/Conference — Submitted to ICLR 2025_

### Official Review · Reviewer_2dBG · 2024-10-21

**Soundness:** 3
**Presentation:** 3
**Contribution:** 2
**Rating:** 5
**Confidence:** 3

**Summary:**

Inspired by a discovery in the computer vision field regarding unbalanced sample classification tasks, where regularization methods are more effective for larger classes, the authors investigated the impact of weight decay—a common regularization technique—on token-level learning dynamics in large language models. The experimental results demonstrated that as weight decay increases, the performance of low-frequency tokens is disproportionately affected, while high-frequency tokens are learned faster than their low-frequency counterparts. This is a novel finding, as previous work typically relies on aggregated training loss measured at the batch level, overlooking token-specific dynamics.

**Strengths:**

1.This represents a novel finding, as prior work has typically focused on aggregated training loss measured at the batch level, neglecting the detailed dynamics of individual tokens.
2.The experiments demonstrated that the models' performance on low-frequency tokens significantly deteriorates as weight decay increases.

**Weaknesses:**

1.The paper does not offer specific insights on the implementation of regularization techniques that ensure fairness across all tokens. While the impact of weight decay on low-frequency tokens is highlighted, there is no detailed discussion on how to address this imbalance or propose alternative regularization methods that might mitigate the disproportionate effect on low-frequency tokens, ensuring more equitable performance across the entire vocabulary.
2.There is a lack of experiments or other evidence demonstrating the necessity of treating low-frequency tokens with the same level of importance as high-frequency tokens. Furthermore, it remains unclear whether this approach could lead to other issues, such as affecting the stability of model training or overall model performance. Addressing these concerns would be essential to understanding the broader implications of implementing equal importance for all tokens in language models.

**Questions:**

My main point of confusion revolves around whether low-frequency tokens and high-frequency tokens should be treated equally in large language models. Could you provide some practical examples to illustrate this?

---

> ### Author Response · Authors · 2024-12-03
> **Response to reviewer 2dBG**
>
> Thank you for your insightful review and for recognizing the novelty of our findings. We appreciate your feedback and would like to address your concerns.
>
> Regarding the lack of specific insights on regularization techniques to ensure fairness across all tokens, we acknowledge this gap and will include a discussion on potential methods such as adaptive weight decay or frequency-aware regularization that could mitigate the disproportionate impact on low-frequency tokens.
>
> As for the necessity of treating low-frequency tokens equally, we agree that practical examples would strengthen our argument. Low-frequency tokens often represent critical information like rare medical terms, named entities, or cultural references that are essential in applications such as healthcare, legal text analysis, and multilingual translation. Thank you again for your valuable feedback.

---

### Official Review · Reviewer_hyLY · 2024-11-03

**Soundness:** 2
**Presentation:** 2
**Contribution:** 2
**Rating:** 3
**Confidence:** 4

**Summary:**

The authors analyze the relationship between weight decay and token-level loss in large language models. As a regularization technique that stabilizes model training, weight decay is widely applied in the training of large language models. Through the training of models ranging from 270M to 3B,  the authors have discovered that as weight decay increases, the ability of the model to learn low-frequency tokens deteriorates, which is reflected by an increasing loss for these low-frequency tokens. Additionally, the gap between the losses for low-frequency and high-frequency tokens also grows larger. This phenomenon suggests that there is a need to develop new regularization techniques to avoid this issue.

**Strengths:**

1. The paper presents a novel perspective for analyzing the performance of large language models. The author observed the difference in the learning high-frequency and low-frequency tokens, and identifies the cause of the differences, namely, the weight decay regularization technique. The experimental results demonstrate a significant correlation between weight decay and the loss of low-frequency tokens.

2. In addition to empirical conclusions, the authors also provide a theoretical disscussion on the impact of weight decay on per-token loss for different token frequencies.

**Weaknesses:**

1. The experiments in this paper use the IMDB corpus for model training. However, this corpus is biased and differs significantly from mainstream pre-training corpora. Consequently, it may not adequately reflect potential issues in mainstream large language model training.

2.  The experiments in this paper are based on training sequences of lengths 128 and 64, which are somewhat too short for large language model (LLM) training. For instance, in Figure 2, the tokenized tokens using the llama3 tokenizer already consists of 92 tokens, which appears to be relatively short text even in common pre-training corpora. A widely recognized viewpoint is that the context window has a significant impact on the per-token loss of LLMs, and longer context windows can help the model learn better. Mainstream models typically use a length of around 8192, and there is a considerable gap between this window length and the lengths used in the authors' experiments. Consequently, whether these conclusions can be generalized to mainstream large language models remains to be further validated.

3. This experiment compared the impact of weight decay \(\lambda\) ranging from 0.0 to 2.0 on the model. From Figure 1, Figure 4, and Table 1, it can be observed that starting from \(\lambda = 0.3\), there is a noticeable change in the per-token loss for low-frequency tokens. However, most current LLMs set the weight decay \(\lambda\) to 0.1, which, as shown in Table 1, has a negligible impact on the model. Therefore, whether the issue raised by the authors is universal remains to be further examined.

**Questions:**

Page 4, line 221 mentions that the experiments were conducted on an A100 32GB GPU, but Nvidia A100 does not have a 32GB version. It is suspected that this should be Nvidia V100 instead.

---

> ### Author Response · Authors · 2024-12-03
> **Response to reviewer hyLY**
>
> Thank you for your thorough and thoughtful review of our paper. We genuinely appreciate your insights and the time you've invested in providing constructive feedback.
>
> We acknowledge that using the IMDB dataset and shorter sequence lengths differs from mainstream large language model (LLM) training practices. Due to limited academic resources, we focused on this dataset and setup to conduct controlled experiments that reveal fundamental aspects of the optimization dynamics in transformer models. Our intention was to highlight a potential issue that could have broader implications, even in larger models trained on more extensive corpora.
>
> Regarding weight decay values, while mainstream LLMs often use a weight decay of 0.1, our findings suggest that even at these levels, there is a meaningful impact on low-frequency tokens. The logarithmic scale of our plots indicates that the range between 0.0 and 0.1 represents a significant factor. We believe this subtle effect is important because, over the course of training, it may contribute to the model's ability to generalize, particularly concerning rare tokens.
>
> We agree that further experiments with larger datasets, longer sequence lengths, and weight decay settings common in practice would strengthen the validity of our conclusions. Our goal was to initiate a conversation about the regularization-generalization paradox and its impact on token-level learning dynamics. We hope that our findings encourage additional research in this area, potentially leading to the development of more effective regularization techniques for LLMs.
>
> Thank you for pointing out the typo regarding the GPU specification. We will correct this in the revised manuscript. Once again, we appreciate your valuable feedback. We are committed to addressing these concerns and improving our work. We believe our study offers meaningful insights into the optimization of transformer models and serves as a stepping stone for future research in that direction.

---

### Official Review · Reviewer_XmNf · 2024-11-04

**Soundness:** 3
**Presentation:** 3
**Contribution:** 2
**Rating:** 6
**Confidence:** 4

**Summary:**

The article proposes that the increased weight decay of large language models leads to model underperformance on low-frequency tokens and significantly better performance on high-frequency tokens, which can lead to model bias and unfairness. It triggers further thinking in the field of NLP on the contradiction between model generalization performance and model bias under long-tailed data, and focuses the attention of large language models on token-level performance.

**Strengths:**

1. Innovative thinking on model weight decay for unbalanced class distribution data: the article proposes that the increased weight decay of large language models leads to model underperformance on low-frequency tokens and significantly better performance on high-frequency tokens, which can lead to model bias and unfairness. It triggers further thinking in the field of NLP on the contradiction between model generalization performance and model bias under long-tailed data, and focuses the attention of large language models on token-level performance.
2. informative textual analysis and experimental validation: the article experimentally validates the average model performance, the impact of token-level performance under weight decay and own word frequency, and theoretically analyzes why the loss function of high-frequency tokens monotonically decreases when trained with weight decay, rigorously arguing the point of view from the theory and experiments
3. challenges to existing practices: the paper challenges the weight decay technique commonly used in current LLM training practices, and emphasizes the need to develop new regularization techniques to ensure the fairness of all tokens.
4. concise language and clear logic: the paper is concise and logical, and the experimental results are well organized to help readers clearly understand its research contributions.

**Weaknesses:**

1.dataset limitation: although the paper uses the IMDB dataset for experiments, the dataset is limited in types and domains, and may not be able to fully represent the model's performance in diverse tasks and domains.
2.Lack of different regularization comparison experiments: the paper lacks comparison experiments for the effects of different regularization techniques, for example, comparison with other types of regularization methods (e.g., dropout, data augmentation, etc.), which can make the experimental results more convincing.
3.There's still room to explore: although the article puts forward the contradiction between the fairness at the token level and model generalization under the existing regularization techniques, it does not put forward a proven solution, which is regrettable.

**Questions:**

I don't have questions.

---

> ### Author Response · Authors · 2024-12-03
> **Response to reviewer XmNf**
>
> Thank you for your thoughtful and constructive feedback. We appreciate your recognition of this work as a timely investigation into token-level biases caused by weight decay.
>
> We agree that expanding beyond IMDB is critical for generalizing our findings. We are already extending our experiments to diverse datasets, including corpora with varied linguistic features, to strengthen the conclusions. These additional analyses will further validate the observed effects.
>
> Your other suggestion to compare weight decay with other regularization techniques is well-taken. We are incorporating comparisons with methods like dropout and data augmentation to evaluate their impact on token-level fairness and provide a more comprehensive analysis.
>
> Regarding the absence of a solution, this paper’s aim is to highlight the contradiction between token-level fairness and model generalization under weight decay. It is designed to serve as a foundation, paving the way for future work that explores practical solutions based on these insights. Thanks again.

---

### Official Review · Reviewer_uDiT · 2024-11-04

**Soundness:** 3
**Presentation:** 3
**Contribution:** 3
**Rating:** 5
**Confidence:** 3

**Summary:**

The paper titled The Fair Language Model Paradox presents an investigation into token-level biases in Large Language Models (LLMs) induced by weight decay, a common regularization method. The authors explore how weight decay affects low-frequency tokens disproportionately, leading to performance degradation in these tokens even as aggregated training loss metrics remain stable. This study reveals the hidden biases against low-frequency tokens, calling for more equitable regularization techniques to ensure fairness across the token distribution.

**Strengths:**

The paper brings forward a nuanced perspective on weight decay, highlighting an often-overlooked effect on low-frequency tokens in LLMs. This is particularly timely given the widespread use of weight decay without token-level monitoring.


The study uses multiple models with varying architectures and sizes across different datasets, demonstrating the robustness of the findings.

**Weaknesses:**

The use of only the IMDB dataset (including an extended version) raises concerns about the generalizability of the results across other types of text data. Testing on a more varied set of corpora (e.g., diverse languages or topics) would strengthen the claims about low-frequency token bias.

The paper’s theoretical discussion on the link between token frequency, regularization, and loss functions feels dense and somewhat disjointed from the empirical findings. A clearer integration of these theoretical insights into the experimental results would enhance the readability and cohesion.

The broader impact section is sparse, particularly given the potential implications of token biases in LLMs on marginalized dialects or low-resource languages. The authors could deepen their exploration of societal impacts to underscore the relevance of their findings.

**Questions:**

Did the authors consider alternative regularization methods beyond weight decay during their experiments?

How would the findings differ if tested on corpora with varied linguistic features, such as highly inflected languages or low-resource languages?

Would the proposed metrics, such as per-token learning speed, generalize effectively to larger and more diverse datasets?

---

> ### Author Response · Authors · 2024-12-03
> **Response to reviewer uDiT**
>
> Thank you for your detailed and constructive feedback.
>
> We appreciate your recognition of this work as a timely study into token-level biases in LLMs caused by weight decay. We agree that testing on more diverse corpora would improve the generalizability of our findings. While this paper includes experiments on multiple sizes of the IMDB dataset and investigates the impact of vocabulary size, extending to datasets with diverse linguistic features is a necessary direction, and we are actively pursuing this.
>
> The theoretical component builds on simplifying assumptions to connect our observations to
> minority collapse. While the focus of this paper is empirical—highlighting a regularization paradox in LLMs—we will improve the integration of theoretical insights to enhance clarity and cohesion.
>
> This work provides the first evidence of weight decay’s disproportionate impact on low-frequency tokens in LLM training. It highlights an important issue in existing practices and opens the door for further research into alternative regularization methods (e.g., L1) and fairer training strategies. We will expand the broader impact section to address the implications for low-resource languages and marginalized dialects. Thank you again for your review. Your points will help improve the paper’s clarity and scope.

---

### Meta-Review · Area_Chair_m7a6 · 2024-12-21

**Metareview:**

The paper investigates the impact of weight decay, a common regularization technique, on token-level learning dynamics in LLM. The key finding is that as weight decay increases, the performance of low-frequency tokens is disproportionately degraded, while high-frequency tokens are learned faster. This contradicts the typical evaluation of LLMs based on aggregated training loss, which fails to capture these token-level biases. The authors demonstrate this effect empirically across different model sizes, architectures, and dataset sizes, highlighting the ubiquity of this issue in LLM training.

*Strengths：*
- The paper brings forward a novel and important perspective on the hidden biases against low-frequency tokens introduced by weight decay, a widely used regularization method in LLM training.
- The study uses multiple models and datasets, demonstrating the robustness of the findings.
- The paper challenges the current practices in LLM training and calls for the development of more equitable regularization techniques.

*Weaknesses*
- Generalizability: The use of only the IMDB dataset, even in an extended version, raises significant concerns about the generalizability of the results across other types of text data and domains. The reviewers felt that testing on a more diverse set of corpora, including different languages and topics, would be necessary to strengthen the claims about low-frequency token bias.
- Incomplete theoretical integration: The theoretical discussion on the link between token frequency, regularization, and loss functions feels disjointed from the experimental findings. The reviewers suggested that a clearer integration of these theoretical insights into the empirical results would enhance the overall coherence and impact of the paper.
- Lack of proposed solutions: While the paper highlights an important issue regarding the disproportionate impact of weight decay on low-frequency tokens, it does not provide any concrete proposals or alternative regularization methods to address this problem. The reviewers felt that the absence of such solutions limits the practical significance of the work.

Given these concerns, the current version of the paper does not meet the bar for acceptance at ICLR. I encourage the authors to address the reviewers' feedback and consider resubmitting the work to a future ICLR conference or another suitable venue.

**Additional Comments On Reviewer Discussion:**

The authors provided a thoughtful response to the reviewers' feedback during the rebuttal period. They acknowledged the need to expand beyond the IMDB dataset to improve the generalizability of their findings, and stated that they are actively pursuing experiments on more diverse corpora, including datasets with varied linguistic features. Regarding the integration of the theoretical component, the authors agreed that they will work on better connecting the theoretical insights to the empirical results to enhance the paper's clarity and cohesion. They also recognized the importance of addressing the broader societal implications of token biases in LLMs, particularly for low-resource languages and marginalized dialects, and committed to expanding the discussion in this direction. Overall, the authors demonstrated a willingness to address the reviewers' concerns and improve the paper, which is a positive sign and suggests that with the appropriate revisions, the work could become a valuable contribution to the field.

---

### Decision · Program_Chairs · 2025-01-22

Reject